# Evolutionary conservation of a core root microbiome across plant phyla along a tropical soil chronosequence

Yun Kit Yeoh [1], Paul G. Dennis[2], Chanyarat Paungfoo-Lonhienne[2], Lui Weber[3], Richard Brackin [2], Mark A. Ragan[4], Susanne Schmidt[2] & Philip Hugenholtz [1]

Culture-independent molecular surveys of plant root microbiomes indicate that soil type generally has a stronger influence on microbial communities than host phylogeny. However, these studies have mostly focussed on model plants and crops. Here, we examine the root microbiomes of multiple plant phyla including lycopods, ferns, gymnosperms, and angiosperms across a soil chronosequence using 16S rRNA gene amplicon profiling. We confirm that soil type is the primary determinant of root-associated bacterial community composition, but also observe a significant correlation with plant phylogeny. A total of 47 bacterial genera are associated with roots relative to bulk soil microbial communities, including well-recognized plant-associated genera such as *Bradyrhizobium, Rhizobium*, and *Burkholderia*, and major uncharacterized lineages such as WPS-2, Ellin329, and FW68. We suggest that these taxa collectively constitute an evolutionarily conserved core root microbiome at this site. This lends support to the inference that a core root microbiome has evolved with terrestrial plants over their 400 million year history.

[1] Australian Centre for Ecogenomics, School of Chemistry and Molecular Biosciences, The University of Queensland, Brisbane, QLD 4072, Australia. [2] School of Agriculture and Food Sciences, The University of Queensland, Brisbane, QLD 4072, Australia. [3] Biodiversity Assessment and Management, 26-40 Delancey Street, Cleveland, QLD 4163, Australia. [4] Institute for Molecular Bioscience, The University of Queensland, Brisbane, QLD 4072, Australia. Correspondence and requests for materials should be addressed to S.S. (email: susanne.schmidt@uq.edu.au) or to P.H. (email: p.hugenholtz@uq.edu.au)

Plant roots harbor limited microbial diversity relative to soil that surrounds them, and are usually dominated by a small number of bacterial lineages. Surveys of root microbiomes associated with angiosperms such as *Arabidopsis thaliana*[1, 2], maize[3], oak[4], barley[5], rice[6], lettuce[7], and sugarcane[8] typically reveal Actinobacteria and Proteobacteria as the dominant phyla, suggesting that certain members of these lineages may be consistently enriched in the plant root environment. However, most existing root microbiome studies are of domesticated plants that may not be representative of native plants[9]. Since plants recruit root microbial communities primarily from the soils they inhabit, soil type is considered one of the key determinants of root microbial community composition[10]. Plant host phylogeny is a secondary factor influencing root microbial community composition, and the effect size appears considerably smaller than soil type as was demonstrated with *Arabidopsis* ecotypes[1, 2]. Surveys comparing root-associated communities in maize, sorghum, and wheat[11] (all monocots), as well as *Arabidopsis* and the related species *Cardamine hirsuta*[12] (eudicots), report greater variation in root community composition between more distantly related plants. These studies hint at a broader influence of host phylogeny on root microbiome composition than is currently appreciated.

Since host effects are subtle in shaping the root microbial community relative to soil type, we predicted that a stronger host effect would be detected when comparing more distantly related plant taxa. To date, small-subunit ribosomal RNA gene (16S) sequencing-based root community surveys have been conducted mostly on angiosperms with a focus on model plants (e.g., *Arabidopsis*, poplar) and crops (e.g., wheat, maize, rice, barley, sugarcane, lettuce, grapevine, oat, and pea). By contrast, the root communities of non-angiosperms are poorly characterized; to our knowledge, only a few studies describe the root community composition of non-seed and non-flowering plants[13–15].

Here, we extend the scope of plant host lineages to non-seed (lycopods and ferns) and seed plant phyla (gymnosperms and angiosperms) through investigation of the root microbiomes of 31 plant species in 25 families and 19 orders. The plant species in our study grow in close proximity to one another along a coastal tropical soil chronosequence that spans ~460,000 years. Sites along this chronosequence span 10 km from the youngest to oldest site and experience negligible differences in climate. The chosen chronosequence has a phylogenetically diverse flora composed of ancient and modern plant lineages, and a considerable overlap of plant species between communities facilitating the goal of distinguishing host and soil determinants of root microbial communities. Our study shows that root bacterial community composition is significantly correlated with host phylogeny despite the stronger effect of soils on these communities. Moreover, a core root microbiome was identified at the study site that comprises both well-known plant-associated taxa and poorly characterized and as yet uncultured taxa.

## Results

**Cooloola study site.** Samples were obtained from a well-characterized chronosequence of coastal sand dunes in the Great Sandy National Park at Cooloola, Queensland, Australia[16, 17]. The chronosequence consists of six dune systems across an ~10 km transect (Fig. 1), each of which harbors phylogenetically diverse flora consisting of lineages that were present on the Australian landmass during Gondwanan times and those whose ancestors dispersed into Australia since it became isolated[18, 19]. The older dunes have developed into giant podzols over a period of 460,000 years. We chose six plant communities across four dune systems (Fig. 1, Supplementary Fig. 1, Supplementary Table 1) that share component plant species in early succession sclerophyll woodlands, mid-succession forests, and

late-succession retrogressive woodland and shrubland[17, 19]. This setting facilitates comparison of root microbial community similarity and plant phylogenetic distance while controlling for soil type. In addition, the proximity of the chosen sites to one another ensured that any effects of soil type and plant phylogeny on the root microbiome were not confounded by differences attributable to climate. With evidence that plant age can affect root microbial communities[20], all sampled plants were mature individuals of perennial or biannual species.

We collected 470 samples (235 root and 235 associated bulk soil) from 31 plant species across six plant communities (Fig. 2). We successfully extracted DNA and amplified 16S rRNA gene amplicon sequences to produce bacterial community profiles for 183 and 225 root and soil samples, respectively. Chimeric sequences were removed and the remaining data were error-corrected, leaving 3,598,535 sequences. These sequences were clustered into 177,758 operational taxonomic units (OTUs) each comprising ≥10 sequences (i.e., no OTUs were represented by less than 10 sequences) at a sequence similarity threshold of 99%, which corresponds approximately to species-level units[21]. Following taxonomic assignment, chloroplast, mitochondrial, and unassigned sequences were removed. Two approaches were then used to normalize for sequencing depth—the first was a centered log ratio normalization with total sum scaling and the other was sequence rarefaction to 1000 reads per sample followed by correction of read counts to adjust for variation in bacterial lineage-specific 16S rRNA gene copy numbers[22]. Low-abundance OTUs were filtered out from the rarefaction-based OTU table by removing those with less than 0.1% relative abundance in any sample.

A range of soil physicochemical characteristics as well as microbial biomass and activity were determined for five replicate bulk soil samples per plant community. Each bulk soil replicate represented a pool of three to ten soil samples. As expected, chemical characteristics of the upper soil horizon varied between sites; however, pH was relatively uniform only ranging between 4.1 and 4.6, which is in the normal range for ~36% of tropical land area worldwide[23]. Higher concentrations of metals (aluminum, chromium, iron, potassium, magnesium, manganese, sodium, nickel, strontium, and zinc) were detected in younger soils as these elements coat the silica sand grains that form the original dune substrate but are lost from the upper soil horizons over time (Fig. 3 and Supplementary Table 2). Higher levels of carbon, nitrogen, calcium, phosphorus, and sulphur were detected in rainforest soils, with carbon and nitrogen enrichment as a result of biological processes, and enrichment of other elements the consequence of plants extracting these nutrients from the deeper soil[24]. The most ancient soils are the most nutrient-depauperate as net nutrient losses occur with repeated wildfires and rainfall leaching nutrients into the deep soil out of reach of roots[17, 19] (Fig. 3). Soil microbial biomass and total enzyme activity peaked in the mid succession rainforest soils (sites c and d), but phosphatase activity remained high in the ancient soils, indicating an increased relative microbial investment in phosphorus acquisition (Fig. 3).

**Bulk soil microbiomes.** In total, 17,429 bacterial OTUs were detected in soil samples across the chronosequence at greater than 0.1% relative abundance in at least one sample, of which only 20.5% were shared between all soils, but these accounted for 76.1% of the average relative abundance. The most abundant soil taxa were members of the *Alphaproteobacteria*, Actinobacteria, and Acidobacteria consistent with previous studies[25, 26]. Collectively, they represented 79.6% of taxa in each soil based on relative abundance (Supplementary Fig. 2).

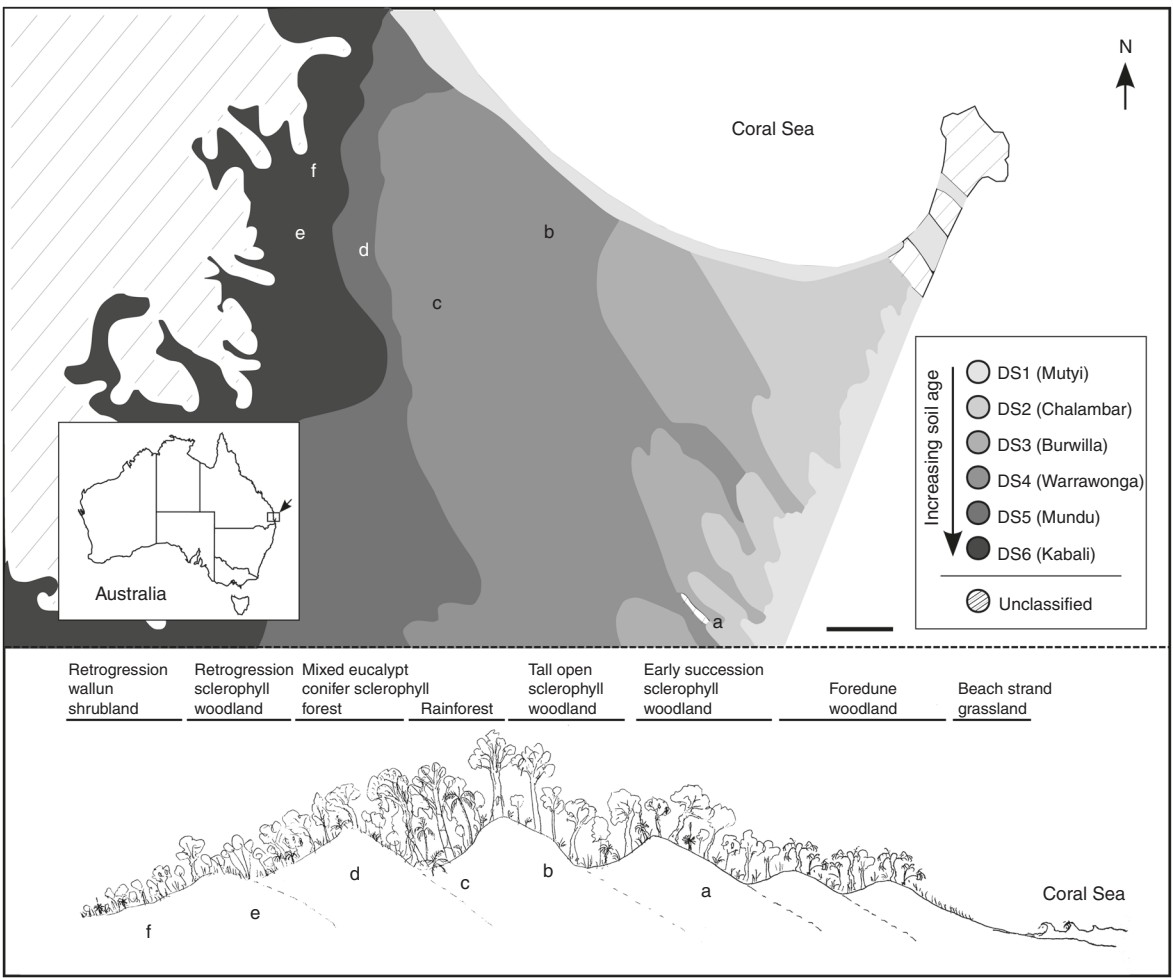

**Fig. 1** Overview of Cooloola study site. Geographical location of the six plant communities sampled (sites a–f) in the Great Sandy National Park at Cooloola, Queensland, Australia, in relation to dune systems and vegetation types. Site a, open sclerophyll *Eucalyptus racemosa* early successional woodland; site b, *Eucalyptus pilularis* tall open moist sclerophyll forest; site c, rainforest (complex notophyll vine forest) with *Agathis robusta*, *Ficus*, and *Archontophoenix cunninghamiana* in fire-sheltered parabolic high dunes; site d, mixed eucalypt conifer open sclerophyll forest with *Eucalyptus racemosa*, *Angophora leiocarpa*, and *Callitris rhomboidea*; site e, retrogression sclerophyll shrubby woodland of *Eucalyptus racemosa*, *Banksia aemula*, and *Leptospermum* species; site f, retrogression Wallum shrubland with *Banksia aemula*, *Xanthorrhoea johnsonii*, and a high diversity of heath shrubs. *Scale bar* represents 1 km

To determine the relative contributions of soil chemical characteristics and plant phylogeny to changes in soil microbial community composition, we used permutational multivariate analysis of variance (PERMANOVA). Firstly, we used principal component analysis (PCA) to summarize the variation in elemental composition (C, N, P, K, S, Mg, Mn, Fe, Al, Zn, Na, Cu, Ni, Ba, Ca, Cr, and Sr) between soils (Supplementary Fig. 3a). This analysis captured a combined 76.3% of variation in soil chemical characteristics between soils in the first two axis scores, which were then used as predictor variables in the PERMANOVA model. Secondly, we used PCA to summarize variation in plant phylogenetic relatedness (Supplementary Fig. 3b) as represented by a distance matrix generated from a multiple sequence alignment of plant ribulose-1,5-bisphosphate carboxylase gene (*rbcL*) sequences (Supplementary Data 1). This analysis captured a combined 83.1% of variation in phylogenetic relatedness between plants in the first two axis scores, which in combination with those representing variation in soil chemical characteristics, were used as predictor variables to explain turnover in soil microbial community composition. Our PERMANOVA model revealed that variation in soil microbial community composition was significantly associated with soil chemical characteristics but not with plant phylogeny (Table 1a). Hierarchical clustering,

PCA, and redundancy analysis (RDA) ordination of the same data gave consistent results in that bulk soil communities predominantly clustered by soil type (Supplementary Figs. 4–6), with the composition of rainforest soil communities being distinct from other soils. Alpha diversity metrics indicated that rainforest soil communities were the most phylogenetically diverse, whereas observed species richness and estimated richness (Chao1) were comparable across all soils (Fig. 4).

**Root-associated microbiomes**. Since the diversity of root microbial communities varies along the longitudinal root axis[27, 28], we isolated DNA only from root apices (five centimeters to the root tip) as this is the primary site of root exudation[29]. A total of 15,991 bacterial OTUs were detected in root samples at greater than 0.1% relative abundance in at least one sample. When compared with bulk soil, root communities had consistently lower species richness and diversity[1, 2, 5] (observed species richness, Chao1, and Faith's phylogenetic diversity; $p < 0.001$, Mann–Whitney $U$-test). This trend included root bacterial communities in the rainforest despite greater phylogenetic diversity of microbial communities in rainforest soil than the other sites (Fig. 4). Diversity metrics of root communities

| Phylum | Order | Family | Species | Plant communities a | b | c | d | e | f |
|---|---|---|---|---|---|---|---|---|---|
| Lycopodiophyta | Lycopodiales | Lycopodiaceae | *Lycopodiella cernua* | | | | | | 3 |
| Pteridophyta (ferns) | Polypodiales | Dennstaedtiaceae | *Pteridium esculentum* | 3 | 2 | | 2 | 1 | 1 |
| | Polypodiales | Polypodiaceae | *Microsorum punctatum* | | | 3 | | | |
| | Schizaeales | Schizaeaceae | *Schizaea bifida* | 2 | | | | | |
| | Schizaeales | Schizaeaceae | *Schizaea dichotoma* | 3 | | 3 | 3 | | 4 |
| Gymnosperms | Cycadales | Zamiaceae | *Macrozamia douglasii* | | | 2 | | 2 | |
| | Pinales | Araucariaceae | *Agathis robusta* | | 1 | 4 | 1 | | |
| | Pinales | Cupressaceae | *Callitris rhomboidea* | | 2 | | 3 | 2 | |
| | Pinales | Pinaceae | *Pinus elliottii* * | | | | 1 | | |
| | Pinales | Podocarpaceae | *Podocarpus elatus* | | | 3 | | | |
| Magnoliid | Canellales | Winteraceae | *Tasmannia insipida* | | | 2 | | | |
| | Laurales | Lauraceae | *Endiandra discolor* | 3 | 3 | 1 | 2 | | |
| | Laurales | Lauraceae | *Endiandra sieberi* | | 3 | | 3 | | 2 |
| | Magnoliales | Annonaceae | *Polyalthia nitidissima* | | | 2 | 1 | | |
| | Magnoliales | Eupomatiaceae | *Eupomatia laurina* | | | 3 | | | |
| | Piperales | Piperaceae | *Piper caninum* | 3 | 3 | 3 | 2 | | |
| Eudicot | Ericales | Ericaceae | *Monotoca sp.* (Fraser Island P. Baxter 777) | 2 | 3 | | 3 | 1 | 3 |
| | Gentianales | Apocynaceae | *Marsdenia rostrata* | 1 | | | | | |
| | Proteales | Proteaceae | *Banksia integrifolia subsp. integrifolia* | 2 | | | 3 | | |
| | Dilleniales | Dilleniaceae | *Hibbertia scandens* | 2 | | | 2 | | |
| | Fabales | Fabaceae | *Acacia disparrima subsp. disparrima* | 4 | | | 3 | | 2 |
| | Fabales | Fabaceae | *Acacia flavescens* | 2 | | | 3 | | 3 |
| | Fabales | Fabaceae | *Daviesia umbellulata* | 1 | | | | | |
| | Fagales | Casuarinaceae | *Allocasuarina littoralis* | 3 | | | 3 | | 3 |
| | Myrtales | Myrtaceae | *Austromyrtus dulcis* | 3 | | | | | |
| Monocot | Asparagales | Asphodelaceae | *Dianella caerulea var. vannata* | 2 | 2 | 2 | | | 1 |
| | Asparagales | Orchidaceae | *Calanthe triplicata* | | | 3 | | | |
| | Liliales | Smilaceae | *Smilax australis* | 3 | 2 | 3 | 3 | | |
| | Liliales | Smilaceae | *Smilax glyciphylla* | 3 | 3 | 2 | 3 | | 3 |
| | Poales | Poaceae | *Imperata cylindrica* | 3 | | | | | |
| | Poales | Poaceae | *Themeda triandra* | 3 | | | | 2 | 1 |

* Naturalized exotic conifer from North America

**Fig. 2** Plant species and number of root samples for which microbial community profiles were successfully obtained. Relationships among the major plant phyla are indicated by a *cladogram* to the *left* of the figure

were also largely comparable between plant orders, except for the eudicot *Dilleniales* (*Hibbertia scandens*) with consistently lower scores relative to other eudicots (Supplementary Fig. 7). Between plant phyla, root communities of the basal lycopod lineage scored, albeit not statistically significant (Kruskal–Wallis test), higher estimated species richness and phylogenetic diversity compared to other phyla (Supplementary Fig. 7). It is possible that these higher values reflect a less selected root microbiome in lycopods before the evolution of root communities with more recent plants. However, additional lycopod root samples collected from different sites are required to verify this hypothesis as we identified lycopods only in one of the Cooloola plant communities (Fig. 2).

At high taxonomic ranks (phylum and class), root bacterial communities were similar to each other and to bulk soils, with communities being dominated by *Alphaproteobacteria* (average 41.7% relative abundance of root-associated community), Actinobacteria (19.1%), and Acidobacteria (17.3%; Supplementary Fig. 2). Despite a lower relative abundance compared to these more dominant lineages, *Betaproteobacteria* were enriched approximately fourfold in roots (average 5.7% relative abundance in root communities) relative to soil (1.4%), possibly indicating selective enrichment in the root environment[1, 2, 6]. This gross similarity between roots and their respective bulk soil communities broadly reflects that root-associated communities are enriched subsets of populations predominantly acquired from the surrounding soil microbiome[1, 2, 30].

At the OTU level, the effect of soil type and host phylogeny on the root bacterial community composition was discernible. For example, hierarchical clustering (Supplementary Fig. 4) and ordination (Supplementary Figs. 5 and 8) of root bacterial communities showed localized groupings by both factors. To determine the relative contributions of soil chemical characteristics and plant phylogeny to changes in root bacterial community composition, we used PERMANOVA as described above for bulk soils. Root bacterial community composition was significantly associated with both soil chemical characteristics and host phylogeny (Table 1b). In contrast, the composition of bulk soil bacterial communities was strongly associated with soil chemical characteristics only (Table 1a). Further support for an association between host phylogeny and root bacterial community composition was provided by Procrustes analysis (correlation = 0.20, $p = 0.02$, number of permutations = 3000) and Mantel test (Spearman $r = 0.11$, $p = 0.018$) that revealed a small but significant correlation between ordinations summarizing variation in root bacterial community composition and plant phylogenetic distance. These findings suggest that within the Cooloola chronosequence, root bacterial communities have evolved in concert with their hosts. Similar studies comparing root communities between the monocot grasses maize, sorghum, and wheat[11], and *Arabidopsis* species and the closely related *Cardamine hirsuta*[12] have also implicated host phylogeny as a contributing factor to root community diversification, albeit secondary to other influences, which may include

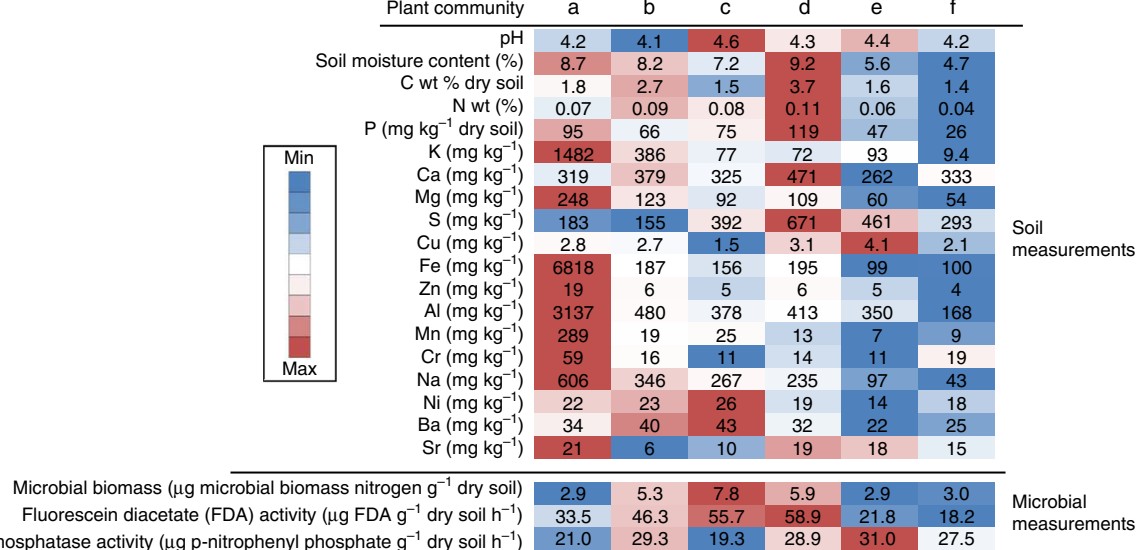

**Fig. 3** Soil chemical characteristics, microbial biomass, total enzyme, and phosphatase activity measured in bulk soils. Values shown are the average of five soil samples from each plant community sampling site, each sample a pool of three to ten individual soils (see Supplementary Table 2 for standard error of the mean and statistical analyses). Color shading represents concentration, biomass, or enzyme activity, and is based on standardized z-scores. Values above the respective means are colored *red*, values below the mean are colored *blue*

**Table 1 Variation in soil and root communities attributable to soil and plant phylogeny**

| | Degrees of freedom | Sum of squares | F model | $R^2$ | p value |
|---|---|---|---|---|---|
| **a** Soil communities (3000 permutations) | | | | | |
| *Soil nutrient* | | | | | |
| Index 1 | 1 | 16,351 | 8.11 | 0.033 | <0.001 |
| Index 2 | 1 | 16,990 | 8.43 | 0.034 | <0.001 |
| Index 1:index 2 | 1 | 15,820 | 7.85 | 0.032 | <0.001 |
| *Host phylogeny* | | | | | |
| Index 1 | 1 | 2044 | 1.01 | 0.004 | 0.368 |
| Index 2 | 1 | 3059 | 1.52 | 0.006 | 0.022 |
| Index 1:index 2 | 1 | 2730 | 1.35 | 0.006 | 0.053 |
| Residuals | 218 | 439,483 | | 0.885 | |
| **b** Root communities (3000 permutations) | | | | | |
| *Soil nutrient* | | | | | |
| Index 1 | 1 | 6564 | 4.18 | 0.021 | <0.001 |
| Index 2 | 1 | 5275 | 3.36 | 0.017 | <0.001 |
| Index 1:index 2 | 1 | 5735 | 3.65 | 0.019 | <0.001 |
| *Host phylogeny* | | | | | |
| Index 1 | 1 | 2830 | 1.80 | 0.009 | 0.004 |
| Index 2 | 1 | 3504 | 2.23 | 0.012 | <0.001 |
| Index 1:index 2 | 1 | 3199 | 2.04 | 0.011 | 0.002 |
| Residuals | 176 | 276,387 | | 0.911 | |

Values in table derived from permutational multivariate analysis of variance (PERMANOVA).

agricultural management, host–microbe, and microbe–microbe interactions[31]. The significant correlation found in the present study between root-associated bacterial communities and host phylogeny across a wide range of plant species suggests that root microbiomes have evolved with their plant hosts at least since the divergence of lycopods ~400 million years ago. Nonetheless, as soil type has a stronger influence on root community composition, root microbiome surveys across multiple plant phyla should be replicated in separate locations to determine whether the influence of host phylogeny on root community composition is consistent across different geographic conditions.

**Cooloola core root microbiome**. The number of Cooloola core root OTUs varies depending on analysis—369 with indicator species analysis[32] (Table 2 and Supplementary Table 3), 302 using a univariate Welch's *t*-test implemented in STAMP[33] (Supplementary Table 4), and 30 with sparse partial least squares discriminant analysis (sPLSDA) implemented in mixOmics[34] (Supplementary Table 5). Nevertheless, core root OTUs from the three analyses completely overlap phylogenetically except for the bacterial genus *Methylovirgula* indicated in STAMP (Supplementary Fig. 9 and Supplementary Table 6), and represent 47 and 40 classifiable and unclassifiable bacterial genera, respectively. The core root OTUs comprise up to 33.2% of root communities based on relative abundance depending on analysis (Supplementary Tables 3–5), some of which are well-known plant–root-associated bacteria, notably *Bradyrhizobium*[35], *Rhizobium*[36], *Burkholderia*[37], and *Azospirillum*[38]. *Bradyrhizobium* and *Rhizobium* are best known as root-nodulating bacteria of legumes, and supply their hosts with biologically fixed nitrogen[35]. The high relative abundance of *Bradyrhizobium* across multiple plant phyla in the present study suggests that their association with non-legumes may be more widespread than previously appreciated[39]. Non-leguminous plants including *Arabidopsis*, corn, and tomato respond to *Bradyrhizobium* nodulation factors using a common molecular mechanism[40], suggesting that this association predates the evolution of legumes within the eudicots.

The genera *Burkholderia* and *Azospirillum* also contain multiple species of recognized root-associated bacteria[41, 42] that have been detected in roots of crops such as lupin[43], maize[44, 45], and sugarcane[46, 47], and are thought to contribute to plant fitness primarily through biological nitrogen fixation[41] and phytohormone production[42]. Other relatively abundant lineages in the core Cooloola root microbiome include *Mycobacterium* and *Rhodoplanes* (Table 2), which have been detected in plant roots[4, 48, 49], but their ecology and function are unknown. Core genera with cultured representatives that are not well recognized in the context of rhizosphere microbiology include *Actinospica*, *Asticcacaulis*, and *Salinispora*, which have been isolated from roots[50], soils[51–53], or marine sediments[54, 55]. The core set also contains taxa belonging to as yet unnamed lineages with no or few isolates, including candidate phylum WPS-2, the alphaproteobacterial

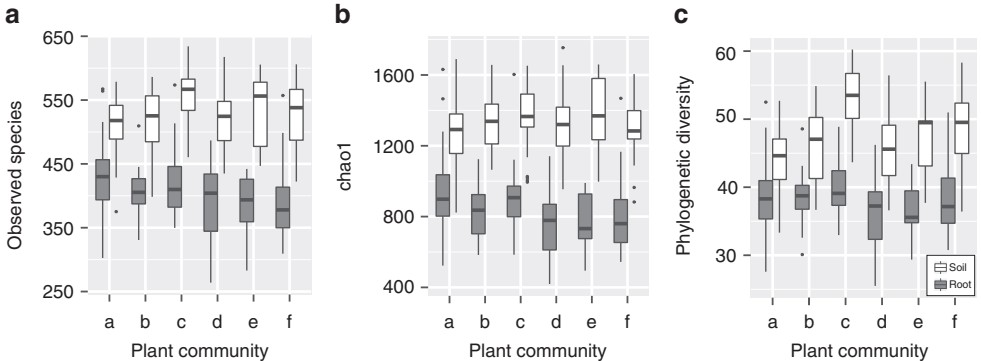

**Fig. 4** Alpha diversity metrics of Cooloola root and bulk soil microbial communities across the six sampling sites. See Fig. 1 for number of samples in each group. *Box* and *whisker plots* showing **a** observed species richness, **b** estimated species richness (chao1), and **c** Faith's phylogenetic diversity. *White rectangles* represent bulk soil communities and *grey rectangles* represent root communities. The centre line within *rectangles* represents the median values, and the two ends of the *rectangles* represent upper and lower quartiles. The *upper whisker* extends to the highest value within 1.5× the interquartile range above the upper quartile, whereas the *lower whisker* extends to the lowest value within 1.5× the interquartile range below the lower quartile. Values outside this range are represented by *black dots*. Total number of root samples is 183 and soil samples is 225

| | | | | | | Average relative abundance (%) | | Also core in: | | | | |
|---|---|---|---|---|---|---|---|---|---|---|---|---|
| **Taxonomy** | | | | | | | | | | | | |
| **Phylum** | **Class** | **Order** | **Family** | **Genus** | **OTUs represented** | **All roots** | **All soils** | **a** | **b** | **c** | **d** | **e** |
| *Actinobacteria* | *Actinobacteria* | *Actinomycetales* | *Actinospicaceae* | *Actinospica* | 2 | 0.40 | 0.04 | | | | | |
| | | | *Actinosynnemataceae* | *Kutzneria* | 1 | 0.09 | 0.03 | | | | | |
| | | | *Mycobacteriaceae* | *Mycobacterium* | 7 | 0.28 | 0.09 | | | | | |
| | | | *Streptomycetaceae* | *Streptomyces* | 2 | 0.06 | 0.02 | Y | Y | Y | Y | Y |
| *Armatimonadetes* | *Fimbriimonadia* | *Fimbriimonadales* | *Fimbriimonadaceae* | *Fimbriimonas* | 1 | 0.06 | 0.02 | Y | | Y | | |
| *Proteobacteria* | *α-proteobacteria* | *Caulobacterales* | *Caulobacteraceae* | *Asticcacaulis* | 3 | 0.25 | 0.03 | Y | | | | |
| | | *Rhizobiales* | *Beijerinckiaceae* | *Methylocapsa* | 2 | 0.15 | 0.04 | | | | | |
| | | | *Bradyrhizobiaceae* | *Afipia* | 1 | 0.24 | 0.00 | Y | | | | |
| | | | | *Bradyrhizobium* | 10 | 5.12 | 3.09 | Y | | | Y | |
| | | | *Hyphomicrobiaceae* | *Rhodoplanes* | 7 | 0.87 | 0.32 | | | | Y | |
| | | | *Phyllobacteriaceae* | *Mesorhizobium* | 1 | 0.12 | 0.03 | Y | Y | Y | | Y |
| | | | | *Nitratireductor* | 1 | 0.04 | 0.00 | | | | | |
| | | | *Rhizobiaceae* | *Agrobacterium* | 1 | 0.16 | 0.01 | Y | Y | Y | Y | Y |
| | | | | *Rhizobium* | 2 | 0.31 | 0.01 | Y | Y | Y | Y | Y |
| | | | | *Shinella* | 1 | 0.09 | 0.00 | | | | | |
| | | | *Xanthobacteraceae* | *Labrys* | 3 | 0.14 | 0.03 | | | | | |
| | | *Rhodospirillales* | *Acetobacteraceae* | *Acidisoma* | 4 | 0.23 | 0.06 | | | | | |
| | | | | *Acidocella* | 2 | 0.19 | 0.01 | | | | | |
| | | | *Rhodospirillaceae* | *Azospirillum* | 6 | 0.54 | 0.01 | | | | | |
| | | | | *Telmatospirillum* | 3 | 0.58 | 0.11 | Y | | | | |
| | | *Sphingomonadales* | *Sphingomonadaceae* | *Sphingomonas* | 1 | 0.10 | 0.02 | Y | Y | Y | Y | Y |
| | *β-proteobacteria* | *Burkholderiales* | *Burkholderiaceae* | *Burkholderia* | 34 | 2.55 | 0.29 | Y | | | | Y |
| | | | | *Salinispora* | 5 | 0.28 | 0.07 | | | | | |
| | | | *Comamonadaceae* | *Rubrivivax* | 2 | 0.07 | 0.00 | | Y | Y | | Y |
| | | | *Oxalobacteraceae* | *Cupriavidus* | 3 | 0.06 | 0.00 | Y | | | | |
| | *γ-proteobacteria* | *Xanthomonadales* | *Xanthomonadaceae* | *Dyella* | 5 | 0.80 | 0.01 | Y | | | | |
| | | | | *Luteibacter* | 1 | 0.05 | 0.00 | Y | | | | |
| | | | | *Rhodanobacter* | 2 | 0.05 | 0.00 | Y | | | | |
| *Tenericutes* | *Mollicutes* | *Anaeroplasmatales* | *Anaeroplasmataceae* | *Asteroleplasma* | 9 | 0.13 | 0.00 | | | | | |

**Table 2 Cooloola core root microbiome taxa summarized by genus classification**

This table of root-associated OTUS is based on indicator species analysis using total sum-scaled relative abundances as input.
"Y" represents "yes" for the five rightmost columns under "Also core in". a: sugarcane[8], Australia; b: *A. thaliana*[1], Germany; c: barley[5], Germany; d: grapevine[60], America; e: maize[3], America.
Please see Supplementary Table 3 for full table showing relative abundances by individual plant phyla and OTUs without genus classification.
Please see Supplementary Tables 4 and 5 for list of root-associated OTUs based on STAMP and sPLSDA, respectively.

order Ellin329, and order FW68 in the phylum Armatimonadetes (Supplementary Table 3). Members of these lineages have been detected in soil habitats[56, 57], but not specifically associated with plant roots. Soil isolates belonging to the order Ellin329 metabolize xylan, arabinose, rhamnose, and starch[58, 59], and are speculated to play a role in plant litter decomposition[59]. Whether these unfamiliar root-associated taxa are actively recruited into the root microbiome, are root-proximal opportunists feeding on rhizodeposits, or interact only with other root-associated bacteria remains to be determined.

**Biogeographical considerations**. Biogeography is often an important factor in shaping the composition of root micro-biomes[1–3, 6, 8]; thus, it is possible that the Cooloola core root microbiome identified in this study (Table 2 and Supplementary Table 3) is specific to the region, or to the continent of Australia. To assess potential biogeographic variation, we cross-referenced the core Cooloola root taxa with root and associated bulk soil microbiome surveys of plants grown in Australia[8] and other countries[1, 3, 5, 60]. We reanalyzed 16S rRNA gene amplicon data from these studies to predict core OTUs via the indicator species method used in the present study for consistency. Several core taxa were shared across multiple studies including *Streptomyces*, *Mesorhizobium*, *Agrobacterium*, *Rhizobium*, *Sphingomonas*, and *Rubrivivax* (Table 2), suggesting that these taxa may be globally important root-associated bacteria. However, other Cooloola core taxa were not identified in the cross-referenced studies, which may indicate regional differences (localized evolution), although methodological variations between the studies (e.g., DNA extraction method) cannot be ruled out as an important contributing factor to compositional differences. It was also noted that the Cooloola data set shared the greatest number of core root taxa (32 of 60, Table 2 and Supplementary Table 3) with Australian sugarcanes[8], possibly reflecting a continental biogeographical signal. From these comparisons we predict that a global core set of plant root microbiota will be considerably more restricted than the list provided in the present study (Table 2 and Supplementary Table 3).

## Conclusion

We identified significant correlation between root community composition and host phylogeny in a survey encompassing plant species from multiple plant phyla growing in close proximity. A core root microbiome dominated by a small number of bacterial taxa was identified. These findings suggest that a core root bacterial community was established before the evolution of modern plant lineages, and root-associated bacterial communities have evolved with their plant hosts. By extension, it is likely that core functionality of the root microbiome is also conserved. Independent root and endosphere metagenome studies have reported a shared functionality relating to traits such as bacterial motility, nitrogen metabolism, iron acquisition, and metabolism, and protein secretion systems in the rhizospheres of rice, cucumber, and wheat[61, 62]. In light of these findings, this study provides a list of bacterial lineages for investigation into their specific plant–microbe interactions including recruitment into the rhizosphere, persistence, function, and turnover, knowledge of which could be used to enhance agricultural crop productivity.

## Methods

**Study site**. An ~10 km transect across a well-characterized coastal dune chronosequence in the Great Sandy National Park (S 25.964, E 153.077) located in Cooloola, south-east Queensland, Australia, was selected as the study site. This location features at least six distinct soil types representing a chronosequence in soil development spanning from young soils several thousand years old to ancient soils ~460,000 years old[16]. The chronosequence exhibits progressive and retrogressive vegetation succession from which we selected multiple plant species representing diverse lineages of the plant kingdom. Climate was identical across the study site thereby minimizing other environmental influences between samples. The rainforest plant community receives similar rainfall but differs in rarity or absence of fire compared to the other fire-prone sclerophyll plant communities.

**Sample collection**. Approval for sample collection at the Great Sandy National Park was obtained from the Queensland Government Department of Environment and Heritage Protection (Permit number: WITK09457411). We sampled the chronosequence in March 2013 after summer rains to obtain a snapshot of the root microbial community composition of 31 plant species. Plants were identified morphologically. Smaller plants (~10–30 cm) were uprooted to access the root system for sampling while larger plants were partially excavated to access roots. Corresponding soil samples were collected from soil (top 10 cm) adjacent to the sampled plant. Where possible, at least three replicate root and soil samples for each plant species were collected. Leaf vouchers were also obtained from each plant sampled. Samples were stored on dry ice in the field and then at −20 °C in the laboratory until further processing.

**Bulk soil nutrient analyses**. Bulk soil samples were pooled into replicates of five according to study site for microbial biomass, activity, and soil chemical compositions. Microbial biomass was measured with chloroform fumigation/extraction followed by a ninhydrin assay for nitrogen content. Microbial activity was assayed by measuring fluorescein diacetate hydrolysis[63]. Soil moisture content was determined gravimetrically (drying at 105 °C for 48 h). Elemental carbon and nitrogen concentrations were measured by combustion in a Dumas apparatus followed by analysis using a LECO TruSpec analyser. Concentrations of other elements were measured by analyzing microwave-digested samples using a Varian Vista Pro inductively coupled plasma optical emission spectrometer.

**Plant phylogeny construction**. Leaf samples approximately 2 × 2 cm were cleaned by dipping in 80% ethanol solution for 1 min followed by washing in sterile water. DNA was extracted from cleaned leaf samples using the PowerSoil® DNA Isolation Kit following manufacturer's instructions. The *rbcL* gene sequence was PCR amplified using primers rbcLa-F 5′-ATGTCACCACAAACAGAGACTAAAGC-3′ and rbcLa-R 5′-GTAAAATCAAGTCCACCRCG-3′. Thermocycling conditions were: 95 °C for 3 min followed by 32 cycles of 95 °C for 30 s, 53 °C for 30 s, 74 °C for 1 min, and finally 74 °C for 10 min. PCR amplicons were cleaned using Agencourt AMPure XP beads (Beckman Coulter Inc.) and capillary sequenced with both forward and reverse primers to obtain a complete amplicon sequence by alignment in Geneious R6[64].

**Microbial DNA extraction and sequencing**. Root tissue up to 3 cm from the root tip was first separated from root samples using a sterile scalpel. Separated root tissues were rinsed with sterile phosphate buffered saline with 0.02% Silwet L-77 surfactant to remove adhering bulk soil particles. DNA was extracted directly from these processed root tissue and soil samples using PowerSoil® DNA Isolation Kits (MO BIO Laboratories, Carlsbad, CA) following manufacturer's instructions. Extracted DNA was quantified using a Qubit fluorometer with Quant-it dsDNA BR assays (Invitrogen™) and then normalized to 4 ng/μl using sterile water. Normalized DNA samples were PCR amplified and sequenced using the 454 GS FLX Titanium pyrosequencing platform. Briefly, 16S rRNA genes were PCR-amplified in 50 μl volumes containing 20 ng DNA, 1X PCR buffer, 0.2 mM of each dNTPs, 1.5 mM MgCl₂, 0.3 mg bovine serum albumin, 0.02 U *Taq* DNA polymerase and 0.2 μM each of primers 27F 5′-AGAGTTTGATCMTGGCTCAG-3′ and 519R 5′-GWATTACCGCGGCKGCTG-3′ modified to contain the 454 FLX Titanium Lib L adapters B and A, respectively. The 519R primer contained a barcode sequence between the primer sequence and adapter. A unique barcode was used to amplify DNA from each sample to facilitate sample identification and demultiplexing after sequencing. Thermocycling conditions were: 95 °C for 5 min followed by 30 cycles of 95 °C for 30 s, 55 °C for 45 s, 72 °C for 90 s; and finally 72 °C for 10 min.

**Sequence data processing for community composition**. Sequence reads were demultiplexed based on their barcode sequences. Adapter, primer and barcode sequences were subsequently removed, and reads were filtered for chimeras using usearch v6.1.544 and corrected for homopolymer errors using Acacia v1.52[65]. Error-corrected sequences were clustered at 99% sequence identity roughly corresponding to species-level units[21] using UCLUST v1.2.22 and cluster representative sequences were assigned taxonomy by BLAST alignment to the Greengenes 16S database[66] (August 2013 release). Chloroplast, mitochondria and low abundance OTUs represented by 10 or fewer sequences in all samples were removed. Sampling depth was rarefied to 1000 reads per sample to calculate alpha diversity metrics. The sequence processing procedures described above were performed using QIIME v1.8.0[67] except for homopolymer error correction using Acacia. Scripts related to the procedures described in this section are provided as Supplementary Software.

**Community diversity and indicator species analyses**. Alpha diversity metrics including observed species richness, Chao1 and Faith's phylogenetic diversity were calculated for all samples using QIIME v1.8.0 based on a rarefied sequence depth of 1000 sequences per sample. For beta diversity analyses, a centered log ratio normalization was first applied to non-normalized OTU sequence counts. Differences in microbial community composition were then visualized using PCA and RDA ordination methods implemented in the R statistical software[68] vegan package[69]. The relative effects of soil type and host phylogeny on root and soil bacterial community composition were assessed using PERMANOVA, available in the vegan package, and principal component scores representing soil chemical characteristics and host phylogeny. Briefly, two distance matrices were constructed, one based on soil chemical characteristics and the other based on plant *rbcL* gene sequence alignments. Soil chemical measurements were first standardized using z-scores, and then principal component scores extracted from PCA performed on the standardized values. Similarly, a distance matrix representing host phylogeny

was constructed using EMBOSS[70] Distmat v6.6.0 (Jin-Nei gamma distance), and principal component scores extracted from PCA performed on this matrix. The latter matrix was constructed using *rbcL* gene sequences amplified from leaf tissue collected during sampling or from public databases if amplification was unsuccessful. Correlation between root/soil community composition and host phylogeny was assessed using the Procrustes and Mantel tests available in the vegan package. Cooloola core root OTUs were determined using indicator species analyses[32] implemented in the R labdsv package[71] on total sum-scaled OTU relative abundances (relative abundance >0.5% in at least one sample) to discriminate between root-associated and soil-associated OTUs. Indicator species analysis was also performed on a rarefaction-normalized OTU table (1000 reads, Supplementary Table 7). The core root community was also assessed using Welch's *t*-test in STAMP v2.1.3[33] and sPLSDA implemented in mixOmics v6.1.1[35] on centered log ratio-transformed OTU counts. R commands are provided as Supplementary Software.

**Root microbiome core taxa comparison with published studies**. External root microbiome survey data sets with associated bulk soil profiles were downloaded from public repositories and processed identically to the Cooloola data set. Indicator OTUs were determined using indicator species analysis in the labdsv package comparing total sum-scaled relative abundances of OTUs in rhizosphere and/or root to soil samples to determine core root taxa.

**Data availability**. The sequence data have been deposited in the NCBI Sequence Read Archive under BioProject accession code PRJNA328519. The authors declare that all other relevant data supporting the findings of the study are available in this article and its Supplementary Information files, or from the corresponding author upon request.

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

## Acknowledgements

We thank Andrew R. Jones for assistance with landscape and dune system classification, Lyn Cook for advice on plant phylogeny and terminology, Maren Westermann, Harshi Gamage, Jitka Kochanek, Scott Buckley, and Deng Yan for processing root samples in the laboratory, Deng Yan also for assistance with sample collection, and David Wood for assistance with statistical analyses. This research was supported through access to facilities managed by Bioplatforms Australia and funded by the Australian Government National Collaborative Research Infrastructure Strategy and Education Investment Fund Super Science Initiative. It was also supported by a Queensland Smart State Co-investment Fund awarded to M.A.R., P.H., and S.S. Y.K.Y. was supported by a University of Queensland International PhD Scholarship.

## Author contributions

Y.K.Y., P.G.D., C.P.-L., S.S., and P.H. designed the experiment. L.W. identified vegetation types and plants at the study site. Y.K.Y., L.W., S.S., and P.H. collected samples, R.B. performed soil microbial and nutrient measurements, and Y.K.Y. and C.P.L. prepared samples for DNA sequencing. Y.K.Y., P.G.D., and P.H. analyzed the data. Y.K.Y., S.S., and P.H. wrote the manuscript, and all authors reviewed drafts of the manuscript.

## Additional information

**Competing interests:** The authors declare no competing financial interests.

