## [Peer Review File · Nature Communications]

Reviewers' comments:

Reviewer #1 (Remarks to the Author):

Yeoh et al have examined variation in root-associate microbial communities (ie, the root microbiome) across six plant communities in Queensland, Australia. They find that soil type is the primary driver of microbial diversity in roots, with a significant secondary impact of plant phylogeny (ie, plant taxa). The group describe a core set of bacteria that are highly enriched on roots compared to bulk soil, and that occur across soil types that encompass a "core root microbiome." Many of these taxa are well known plant associated soilbacteria, such as different groups of rhizobia and other nitrogen fixing plants. This work is some of the first work to describe microbiomes of non-model, non-agricultural plants, forming a basis for future experimental work on microbiomes, and will likely remain significant in the field for many years as a benchmark of root microbiome diversity in natural systems.

The study of microbiomes is still quite new, as the technology needed to accurately describe diverse, unculturable microbial communities has only existed for a few years. This study involves sampling of six soil types, using appropriate techniques to sample bulk soil and root tips of a diverse range of host taxa. Although the number of samples per plant host taxa is fairly low, the overall sampling across soil types is well replicated. The statistical approach is rigorous, involving PERMANOVA to cluster variation across soil types and plant hosts, and then analyzing more carefully OTUs conserved across roots on different soils to assess a core microbiome. The group finds a large core microbiome, with many taxa similar to those found on roots in other regions and on other plant taxa. This is an interesting approach in setting a baseline of which taxa are present. An interesting pattern is the abundance of rhizobial taxa, which are well known as symbionts of legumes but rarely thought of as significant on other plants. Their presence here suggests they may associate at least loosely with a broader set of hosts. Another interesting pattern, albeit one with minimal statistical support, is that the basal land plants appear to have more diverse microbiomes, suggesting they do less to filter soil communities associating with their roots.

This work lays forth several next steps. One next step is clearly an experimental approach to separate more clearly whether microbial community differences are due to the soils or the association of plant taxa with the soils. Although some plant taxa occur across soil types, the plant communities are sufficiently distinct to be categorized as different. Although a rigorous baseline description is necessary, experimentation also clearly has a place to understand the mechanisms of community assembly, persistence, turnover, and of critical importance, microbial function.

Another needed step is implementation of such surveys on a broader scale. This is a pretty well implemented survey, but struggled at broader extrapolation in part due to other datasets not being fully comparable. We need global microbiome studies to better understand microbial biogeography, variation in soil types and climates, etc. But this is a good early step.

The manuscript is well written, easy to follow, and to the best of my observation, free of typos.

Reviewer #2 (Remarks to the Author):

The authors analysed the root microbiomes of the natural vegetation in a coastal ecosystem of Australia using 16S rRNA gene amplicon profiling. The plant species studied (including lycophytes, ferns, gymnosperms and angiosperms) across a soil chronosequence resulted in a root core microbiome of 959 bacterial taxa, which included well-recognized plant-associated genera such as Bradyrhizobium, Rhizobium and Burkholderia, and major uncharacterized lineages.

To study plant-associated microbial diversity associated with Australia's unique plant diversity, is an important objective because the majority especially of the natural vegetation is still unknown. However, due to the facts below I suggest a re-interpretation/writing of the manuscripts with focus on ecology and evolution.

To reduce the study to the question about the impact of soil type / plant genotype on the rhizosphere (which is so manifold studied) is in my eyes to simple. As shown in the results, data only confirm the impact of both, and unfortunately this is not enough novelty to publish it in Nature communication (see this old review: Plant species and soil type cooperatively shape the structure and function of microbial communities in the rhizosphere. Berg G, Smalla K. FEMS Microbiol Ecol. 2009 Apr;68(1):1-13.) However, as the title and last sentence of the abstract indicate there is much more potential within the data.

Other comments:

Abstract. Sentence1: It more difficult, it depends on both but extend is different, e.g. very pronounced in medicinal plant with antimicrobial metabolites.

Abstract. Sentence1: this is not correct; there are a lot of studies on natural plant species and the natural vegetation.

(only some examples: The core microbiome bonds the Alpine bog vegetation to a transkingdom metacommunity. Bragina A, Berg C, Berg G. Mol Ecol. 2015 Sep;24(18):4795-807.; Relating belowground microbial composition to the taxonomic, phylogenetic, and functional trait distributions of trees in a tropical forest. Barberán A, McGuire KL, Wolf JA, Jones FA, Wright SJ, Turner BL, Essene A, Hubbell SP, Faircloth BC, Fierer N. Ecol Lett. 2015 Dec;18(12):1397-405.; Vertical transmission explains the specific Burkholderia pattern in Sphagnum mosses at multi-geographic scale. Bragina A, Cardinale M, Berg C, Berg G. Front Microbiol. 2013 Dec 18;4:394.)

32-39: An introduction with Arabidopsis (partly in pot experiments) and crops is not appropriate for this study. In addition, domestication strongly shaped the crop associated microbiota (more fast growing bacteria, r-strategy), which should be considered.

(Plant domestication and the assembly of bacterial and fungal communities associated with strains of the common sunflower, Helianthus annuus. Leff JW, Lynch RC, Kane NC, Fierer N.

New Phytol. 2016 Nov 23. Impact of plant domestication on rhizosphere microbiome assembly and functions. Pérez-Jaramillo JE, Mendes R, Raaijmakers JM. Plant Mol Biol. 2016 Apr;90(6):635-44.)

66: The rhizosphere was described already in 1901 by Lorenz Hiltner and the similarities with the gut microbiome are here not appropriate.

Recent studies about the seed microbiota is not considered.

(Taxonomic and functional diversity of cultured seed associated microbes of the cucurbit family. Khalaf EM, Raizada MN. BMC Microbiol. 2016 Jun 27;16(1):131.

Klaedtke, S., Jacques, M. A., Raggi, L., Prévieux, A., Bonneau, S., Negri, V., Chable, V., and Barret, M. 2016. Terroir is a key driver of seed-associated microbial assemblages. Environ. Microbiol. 18: 1792-1804.)

Reviewer #3 (Remarks to the Author):

Assessment of Kit Yeoh, et al., "Evolutionary conservation of a core root microbiome across plant phyla along a tropical soil chronosequence"

In this manuscript, the authors examine the root microbiota of a taxonomically varied panel of plant hosts, which include flowering and lower non-seed producing plants such as mosses (Lycopods) and ferns. Although this study is limited to the use of conventional bacterial 16S rRNA amplicon profiling, the experiments were designed and conducted in a unique ecological context. By sampling plants from a wide taxonomic background that grow in close proximity to each other along a coastal tropical soil 'chronosequence', the authors are able to examine the effect of host phylogeny largely unaffected by the confounding effects of climate and biogeography. The authors observe that root microbiota composition is significantly correlated to host phylogeny as well as the presence of a conserved core root microbiome that encompasses the major lineages of land plants that diverged more than 400 million years ago. These findings are presented and discussed in a measured and nuanced manner in the context of our current understanding of the root microbiota. However, there are a number of concerns, mostly regarding the computational analysis performed that need to be addressed and that are listed below.

Main points:

1. All sites sampled are located in a relatively small area, making it impossible to generalize and extrapolate these findings to other soil types and locations. The authors correctly comment on this caveat (l. 231 onwards) and it remains perhaps the most relevant shortcoming of these experiments. This could be particularly problematic given the fact that all sites in the chronosequence have soils unusually acidic (i.e. 4.1 to 4.2), placing them as clear outliers compared to previously tested soils in other root microbiota studies. This might have profound effects since copious evidence indicates that soil pH is one of the most important determinants of soil biota composition. This limitation must be explicitly stated and discussed in the main text.
2. Another point of concern is the choice of computational tools employed to process the amplicon data. A number of significant shortcomings come to mind: how was quality filtering of reads performed (if any)? What OTU clustering algorithm among those implemented in QIIME was employed? How was chimera removal of OTU representative sequences conducted (if at all)? These questions require at the very least detailed clarification and ideally a re-processing of the data using state-of-the-art practices and tools. Quality filtering, chimera removal and a robust OTU clustering algorithm (e.g. UPARSE; Edgar, Nat. Meth., 2013) are exceedingly important. For example, it has been recently shown that subsampling (or rarefaction) of sequences to an even sample depth (in this case 1,000 reads) is not optimal and that even simple normalization schemes (e.g. total sum) are preferable (see McMurdie et al., PloS Comp. Biol., 2013). Finally, I have not been able to find a reference in the manuscript to a public code repository to access the scripts employed in the various analyses. In my opinion, providing well documented scripts is particularly important for microbiome studies in order to facilitate reproducibility. Without a much more detailed description of the methodology or access to the code it is virtually impossible to replicate the results.
3. Further, why was the analysis performed at a resolution of 99% sequence identity instead of the more widely used (and arguably arbitrary) 97% threshold?
4. Have the authors considered using, in addition to the indicator species analysis employed, an arguably more sophisticated (and robust) feature extraction method, such as Random Forests? This approach has been successfully employed in recent microbiota studies (e.g. Dey et al., Cell, 2015). It might be a good idea to test at least one additional algorithm in order to determine the robustness of the reported root core microbiome.
5. Why there is no combined analysis of beta-diversity that allows direct comparison between root samples from all species and soil samples? The closest to this are separate hierarchical clustering dendrograms for soil and root samples (Fig. S4) and a single RDA plot for soil samples (Fig. S5). The fact that this last figure does not contain an additional panel for root samples is perhaps a

mistake since it seems to be referenced in the text (l. 174). I believe it is relevant to provide a general overview of the data and a joint analysis of diversity for all samples, either using RDA, PCoA, NMDS or any similar approach. If technical factors are a problem (e.g. run-to-run variation), this can be addressed using other analytical methods such as CAP analysis. Without a general beta-diversity figure, some claims made in the text are difficult to assess, e.g., the hypothesis that roots of Lycopods constitute a less selective environment from the soil community compared to the other species.

6. Aside from determining conserved microbiota members and assessing their specific weight to the root communities (~33% in this case), it will be also interesting to assess lineage-specific OTUs and their aggregated relative abundance. In other words: are there bacterial taxa which are specific of certain plant species or clades? If so, how much of the root community is determined by these? I believe that indicator analysis can also be used for this purpose.

7. Plant age is known to affect root microbiome profiles (Dombrowski et al 2016 ISME J). Can the authors estimate to what extent hugely varying plant age of their root samples introduces a bias in the definition of the core microbiome? Similarly, given the fact that the present study provides a snapshot of a single sampling survey only, can the authors estimate how seasonal variation in microbiomes might affect the definition of a core microbiome?

Minor points:

1. The authors use a distance matrix from the alignment of the rbcL generalization to obtain distances between host species that represent phylogenetic relatedness. However, the appropriate measure would be tree distances after inference of a species phylogeny using said alignment and incorporating a model of DNA evolution. Although I do not necessarily expect that this will change significantly the results, it might improve the signal observed. Incidentally, there are a number of things that can be done by mapping features of root microbiota on the species tree, for example alpha-diversity, to determine if more basal plant species harbor more complex bacterial communities in general.

2. In the text, the authors mention the results of a Procrustes analysis (l. 180) to further support correlation between host phylogeny and root microbiota. Is there a figure that summarizes this analysis? Additionally, a test of matrix correlation (e.g. Mantel test) would perhaps be an alternative to this analysis.

3. A common assumption in establishing chronosequences is that no other variable besides age has changed between sites of interest. I think this popular ecology approach has substantial limitations because its key assumption, i.e. space-for-time substitution, runs the danger of oversimplification, especially in the context of dune successional sequences described here (see Johnson and Miyanashi, Ecology Letters 2008). Although this is unlikely to affect the overall conclusions of the authors, inherent limitations of this approach in the present dune successional sequence need to be clearly spelled out in the main text.

4. l. 124 'Principle Component Analysis (PCA)' instead of 'Principal'

Reviewers' comments:

Reviewer #1 (Remarks to the Author):

Yeoh et al have examined variation in root-associate microbial communities (ie, the root microbiome) across six plant communities in Queensland, Australia. They find that soil type is the primary driver of microbial diversity in roots, with a significant secondary impact of plant phylogeny (ie, plant taxa). The group describe a core set of bacteria that are highly enriched on roots compared to bulk soil, and that occur across soil types that encompass a "core root microbiome." Many of these taxa are well known plant associated soil bacteria, such as different groups of rhizobia and other nitrogen fixing plants. This work is some of the first work to describe microbiomes of non-model, non-agricultural plants, forming a basis for future experimental work on microbiomes, and will likely remain significant in the field for many years as a benchmark of root microbiome diversity in natural systems.

The study of microbiomes is still quite new, as the technology needed to accurately describe diverse, unculturable microbial communities has only existed for a few years. This study involves sampling of six soil types, using appropriate techniques to sample bulk soil and root tips of a diverse range of host taxa. Although the number of samples per plant host taxa is fairly low, the overall sampling across soil types is well replicated. The statistical approach is rigorous, involving PERMANOVA to cluster variation across soil types and plant hosts, and then analyzing more carefully OTUs conserved across roots on different soils to assess a core microbiome. The group finds a large core microbiome, with many taxa similar to those found on roots in other regions and on other plant taxa. This is an interesting approach in setting a baseline of which taxa are present. An interesting pattern is the abundance of rhizobial taxa, which are well known as symbionts of legumes but rarely thought of as significant on other plants. Their presence here suggests they may associate at least loosely with a broader set of hosts. Another interesting pattern, albeit one with minimal statistical support, is that the basal land plants appear to have more diverse microbiomes, suggesting they do less to filter soil communities associating with their roots.

This work lays forth several next steps. One next step is clearly an experimental approach to separate more clearly whether microbial community differences are due to the soils or the association of plant taxa with the soils. Although some plant taxa occur across soil types, the plant communities are sufficiently distinct to be categorized as different. Although a rigorous baseline description is necessary, experimentation also clearly has a place to understand the mechanisms of community assembly, persistence, turnover, and of critical importance, microbial function.

Another needed step is implementation of such surveys on a broader scale. This is a pretty well implemented survey, but struggled at broader extrapolation in part due to other datasets not being fully comparable. We need global microbiome studies to better understand microbial biogeography, variation in soil types and climates, etc. But this is a good early step.

The manuscript is well written, easy to follow, and to the best of my observation, free of typos.

We thank the reviewer for the positive feedback, and we agree with his/her suggested next steps. We have added sentences at the end of the manuscript to incorporate elements of these next steps (lines 265 - 270).

Reviewer #2 (Remarks to the Author):

The authors analysed the root microbiomes of the natural vegetation in a coastal ecosystem of Australia using 16S rRNA gene amplicon profiling. The plant species studied (including lycophytes, ferns, gymnosperms and angiosperms) across a soil chronosequence resulted in a root core microbiome of 959 bacterial taxa, which included well-recognized plant-associated genera such as Bradyrhizobium, Rhizobium and Burkholderia, and major uncharacterized lineages. To study plant-associated microbial diversity associated with Australia's unique plant diversity, is an important objective because the majority especially of the natural vegetation is still unknown. However, due to the facts below I suggest a re-interpretation/writing of the manuscripts with focus on ecology and evolution.

To reduce the study to the question about the impact of soil type / plant genotype on the rhizosphere (which is so manifold studied) is in my eyes too simple. As shown in the results, data only confirm the impact of both, and unfortunately this is not enough novelty to publish it in Nature communication (see this old review: Plant species and soil type cooperatively shape the structure and function of microbial communities in the rhizosphere. Berg G, Smalla K. FEMS Microbiol Ecol. 2009 Apr;68(1):1-13.) However, as the title and last sentence of the abstract indicate there is much more potential within the data.

We thank the reviewer for recognizing the value of the study. However, we feel that re-interpretation and major rewriting of the ms is unwarranted, as i) this is the first study to assess such a broad plant host range, so a discussion of the relative effects of soil type vs host phylogeny is appropriate despite the topic having been broached in previous publications, and ii) the study is not reduced simply to impact of soil type/plant genotype on the rhizosphere, as the core microbiome analysis is discussed at length in the current ms, not simply alluded to in the title and last sentence of the abstract.

Other comments:

Abstract. Sentence1: It more difficult, it depends on both but extend is different, e.g. very pronounced in medicinal plant with antimicrobial metabolites.

We have added the word “generally” to this sentence to indicate exceptions to this rule of thumb.

Abstract. Sentence2: this is not correct; there are a lot of studies on natural plant species and the natural vegetation. (only some examples: The core microbiome bonds the Alpine bog vegetation to a transkingdom metacommunity. Bragina A, Berg C, Berg G. Mol Ecol. 2015 Sep;24(18):4795-807.; Relating belowground microbial composition to the taxonomic, phylogenetic, and functional trait distributions of trees in a tropical forest. Barberán A, McGuire KL, Wolf JA, Jones FA, Wright SJ, Turner BL, Essene A, Hubbell SP, Faircloth BC, Fierer N. Ecol Lett. 2015 Dec;18(12):1397-405.; Vertical transmission explains the specific Burkholderia pattern in Sphagnum mosses at multi-geographic scale. Bragina A, Cardinale M, Berg C, Berg G. Front Microbiol. 2013 Dec 18;4:394.)

We thank the reviewer for highlighting this additional literature. In an attempt to quantitatively address this issue, we identified 60 studies using culture-independent methods to analyse root microbiomes, based on keyword searches of the Web of Science (keywords: root microbiome 16S). Of these, only 10 (~17%) concerned non-model, non-crop plant species. Therefore we have amended the second sentence of the abstract to read “mostly focussed on model plants and crops”.

32-39: An introduction with Arabidopsis (partly in pot experiments) and crops is not appropriate for this study. In addition, domestication strongly shaped the crop associated microbiota (more fast growing bacteria, r-strategy), which should be considered. (Plant domestication and the assembly of bacterial and fungal communities associated with strains of the common sunflower, Helianthus annuus. Leff JW, Lynch RC, Kane NC, Fierer N. New Phytol. 2016 Nov 23. Impact of plant domestication on rhizosphere microbiome assembly and functions. Pérez-Jaramillo JE, Mendes R, Raaijmakers JM. Plant Mol Biol. 2016 Apr;90(6):635-44.)

Potential impact of domestication on the root microbiome is a good point, which also strengthens the justification for our study of native plants. We have added a caveat sentence to the introduction to highlight this potential skewing of existing root microbiome data (lines 36-38).

66: The rhizosphere was described already in 1901 by Lorenz Hiltner and the similarities with the gut microbiome are here not appropriate. Recent studies about the seed microbiota is not considered. (Taxonomic and functional diversity of cultured seed associated microbes of the cucurbit family. Khalaf EM, Raizada MN. BMC Microbiol. 2016 Jun 27;16(1):131. Klaedtke, S., Jacques, M. A., Raggi, L., Préveaux, A., Bonneau, S., Negri, V., Chable, V., and Barret, M. 2016. Terroir is a key driver of seed-associated microbial assemblages. Environ. Microbiol. 18: 1792-1804.)

We are not making a statement on the precedence of describing the rhizosphere, simply noting the interesting comparison to the animal gut made by others [refs 16,17]. We feel this is appropriate in the context of making the point that vertical transmission of rhizosphere microbiota is more limited than gut microbiota transmission in animals. Regarding vertical transmission through seeds, we have amended the relevant sentence in the text to acknowledge this point (lines 71-72), although the general consensus is that a minority of rhizosphere bacteria are transmitted via this route (Truyens S, Weyens N, Cuypers A and Vangronsveld J. *Environmental Microbiology Reports*. 2015 7(1):40-50).

Reviewer #3 (Remarks to the Author):

Assessment of Kit Yeoh, et al., “Evolutionary conservation of a core root microbiome across plant phyla along a tropical soil chronosequence”

In this manuscript, the authors examine the root microbiota of a taxonomically varied panel of plant hosts, which include flowering and lower non-seed producing plants such as mosses (Lycopods) and ferns. Although this study is limited to the use of conventional bacterial 16S rRNA amplicon profiling, the experiments were designed and conducted in a unique ecological context.

By sampling plants from a wide taxonomic background that grow in close proximity to each other along a coastal tropical soil 'chronosequence', the authors are able to examine the effect of host phylogeny largely unaffected by the confounding effects of climate and biogeography. The authors observe that root microbiota composition is significantly correlated to host phylogeny as well as the presence of a conserved core root microbiome that encompasses the major lineages of land plants that diverged more than 400 million years ago. These findings are presented and discussed in a measured and nuanced manner in the context of our current understanding of the root microbiota. However, there are a number of concerns, mostly regarding the computational analysis performed that need to be addressed and that are listed below.

Main points:

1. All sites sampled are located in a relatively small area, making it impossible to generalize and extrapolate these findings to other soil types and locations. The authors correctly comment on this caveat (l. 231 onwards) and it remains perhaps the most relevant shortcoming of these experiments. This could be particularly problematic given the fact that all sites in the chronosequence have soils unusually acidic (i.e. 4.1 to 4.2), placing them as clear outliers compared to previously tested soils in other root microbiota studies. This might have profound effects since copious evidence indicates that soil pH is one the most important determinants of soil biota composition. This limitation must be explicitly stated and discussed in the main text.

We respectfully disagree with the reviewer that our soil is an outlier based on pH. An estimated 36% of tropical land area worldwide has acidic pHs ~4 (Sanchez PA and Logan TJ. *SSSA Special Publication*. 2012 29: 35-46). However, we have amended the relevant sentence to note this explicitly in the text (line 110-111).

2. Another point of concern is the choice of computational tools employed to process the amplicon data. A number of significant shortcomings come to mind: how was quality filtering of reads performed (if any)? What OTU clustering algorithm among those implemented in QIIME was employed? How was chimera removal of OTU representative sequences conducted (if at all)? These questions require at the very least detailed clarification and ideally a re-processing of the data using state-of-the-art practices and tools. Quality filtering, chimera removal and a robust OTU clustering algorithm (e.g. UPARSE; Edgar, Nat. Meth., 2013) are exceedingly important. For example, it has been recently shown that subsampling (or rarefaction) of sequences to an even sample depth (in this case 1,000 reads) is not optimal and that even simple normalization schemes (e.g. total sum) are preferable (see McMurdie et al., PloS Comp. Biol., 2013). Finally, I have not been able to find a reference in the manuscript to a public code repository to access the scripts employed in the various analyses. In my opinion, providing well documented scripts is particularly important for microbiome studies in order to facilitate reproducibility. Without a much more detailed description of the methodology or access to the code it is virtually impossible to replicate the results.

We thank the reviewer for pointing out this missing information. Procedures regarding software used and their respective version numbers have been added to the methods section (lines 335-345).

These include usearch6.1 for chimera filtering, Acacia for sequence homopolymer error correction and UCLUST for sequence clustering.

In accordance with the reviewer's suggestion, we have re-processed the data using total sum scaling and centred log ratios instead of rarefaction to 1000 reads per sample, and results, discussion and corresponding figures have been updated accordingly. This approach substantially changed the number of predicted core root OTUs (959 to 369), however, the core root taxa consolidated at genus and higher ranks (Table 2) derived from rarefaction and total sum scaling were consistent. We have moved figures of the rarefaction-based analyses into supplementary material. We have also included a section in supplementary material listing all scripts used in the analysis of these data including commands for sequence processing and statistical analyses.

3. Further, why was the analysis performed at a resolution of 99% sequence identity instead of the more widely used (and arguably arbitrary) 97% threshold?

It is actually preferable to use higher sequence identity thresholds to prevent potential loss of resolution (i.e. amalgamation of distinct species; Patin et al., 2013), however, this comes at the cost of increased noise due to sequencing error, which is why we suggested a 97% identity clustering threshold in 2010 (Kunin et al., 2010). The major type of sequencing error in pyrosequencing data is homopolymers, which can effectively be corrected using Acacia (Bragg et al., 2012), which has allowed us to increase the clustering threshold. We did not enter into these detailed methodological discussions in the manuscript, but could include them in supplementary material if necessary.

4. Have the authors considered using, in addition to the indicator species analysis employed, an arguably more sophisticated (and robust) feature extraction method, such as Random Forests? This approach has been successfully employed in recent microbiota studies (e.g. Dey et al., Cell, 2015). It might be a good idea to test at least one additional algorithm in order to determine the robustness of the reported root core microbiome.

We do not have any experience with Random Forests, so instead have repeated the core root OTU analysis using two related approaches that we are familiar with. The first is a univariate analysis implemented in STAMP (Parks et al., Bioinformatics 2014; 30(21): 3123-3124) and the second is a partial least squares multivariate analysis implemented in mixOmics (Lê Cao et al., BMC Bioinformatics 2011; 12:253). The outputs from these analyses are consistent with the indicator species analysis, thus we have retained the total sum scaled indicator species analysis (Table 2) in the main text and included the additional analyses in supplementary material (Tables S3 and S4). The phylogenetic overlap in core root taxa predicted by these three approaches is around 85%, and is shown in Fig. S9 and Table S5.

5. Why there is no combined analysis of beta-diversity that allows direct comparison between root samples from all species and soil samples? The closest to this are separate hierarchical clustering dendrograms for soil and root samples (Fig. S4) and a single RDA plot for soil samples (Fig. S5).

The fact that this last figure does not contain an additional panel for root samples is perhaps a mistake since it seems to be referenced in the text (l. 174). I believe it is relevant to provide a general overview of the data and a joint analysis of diversity for all samples, either using RDA, PCoA, NMDS or any similar approach. If technical factors are a problem (e.g. run-to-run variation), this can be addressed using other analytical methods such as CAP analysis. Without a general beta-diversity figure, some claims made in the text are difficult to assess, e.g., the hypothesis that roots of Lycopods constitute a less selective environment from the soil community compared to the other species.

We have included a combined root and soil beta diversity analysis in supplementary material (Fig. S5) and corrected the figure referencing to point to the PCA of root communities (Fig. S8).

6. Aside from determining conserved microbiota members and assessing their specific weight to the root communities (~33% in this case), it will be also interesting to assess lineage-specific OTUs and their aggregated relative abundance. In other words: are there bacterial taxa which are specific of certain plant species or clades? If so, how much of the root community is determined by these? I believe that indicator analysis can also be used for this purpose.

We thank the reviewer for this suggestion. We actually did perform this analysis originally, but decided not to include the results in the manuscript due to the high level of noise most likely due to a combination of limited sampling per plant host and complexity of the overall habitat.

7. Plant age is known to affect root microbiome profiles (Dombrowski et al 2016 ISME J). Can the authors estimate to what extent hugely varying plant age of their root samples introduces a bias in the definition of the core microbiome? Similarly, given the fact that the present study provides a snapshot of a single sampling survey only, can the authors estimate how seasonal variation in microbiomes might affect the definition of a core microbiome?

We are aware of the effect of plant age, and endeavoured to only sample mature individuals of perennial or biannual species. We have added a sentence to the methods to highlight this strategy (lines 89-90). We also acknowledge that the study is not sufficiently comprehensive to address all variables, including seasonal variation, due to logistical limitations, but feel that it is a sound basis for subsequent comparative studies.

Minor points:

1. The authors use a distance matrix from the alignment of the rbcL generalization to obtain distances between host species that represent phylogenetic relatedness. However, the appropriate measure would be tree distances after inference of a species phylogeny using said alignment and incorporating a model of DNA evolution. Although I do not necessarily expect that this will change significantly the results, it might improve the signal observed. Incidentally, there are a number of things that can be done by mapping features of root microbiota on the species tree, for example

alpha-diversity, to determine if more basal plant species harbor more complex bacterial communities in general.

We did in fact use corrected distances for this analysis using a gamma distribution model (Jin-Nei, 1990), which is now stated explicitly in the methods (line 361).

2. In the text, the authors mention the results of a Procrustes analysis (l. 180) to further support correlation between host phylogeny and root microbiota. Is there a figure that summarizes this analysis? Additionally, a test of matrix correlation (e.g. Mantel test) would perhaps be an alternative to this analysis.

We performed a Mantel test as suggested and have included the results in the main text (line 188-189). We found the figure of the Procrustes analysis to be uninformative and hence excluded the figure. It is reproduced below for the benefit of the reviewer.

3. A common assumption in establishing chronosequences is that no other variable besides age has changed between sites of interest. I think this popular ecology approach has substantial limitations because its key assumption, i.e. space-for-time substitution, runs the danger of oversimplification, especially in the context of dune successional sequences described here (see Johnson and Miyanashi, Ecology Letters 2008). Although this is unlikely to affect the overall conclusions of the authors, inherent limitations of this approach in the present dune successional sequence need to be clearly spelled out in the main text.

We are not making the space-for-time substitution argument despite using the term chronosequence in the manuscript. Rather the site is useful for the study purely from the standpoint of having distinct soil types in close proximity with overlapping vegetation.

4. 1. 124 'Principle Component Analysis (PCA)' instead of 'Principal'

We believe the correct spelling is Principal Component Analysis (https://en.wikipedia.org/wiki/Principal_component_analysis)

REVIEWERS' COMMENTS:

Reviewer #2 (Remarks to the Author):

[No further comments for author.]

Reviewer #3 (Remarks to the Author):

Revised manuscript by Yeoh, et al., "Evolutionary conservation of a core root microbiome across plant phyla along a tropical soil chronosequence".

The authors have adequately addressed my major concerns, including the question of how generalizable these findings are in light of the acidic soils of the chronosequence. The revised version of the manuscript now incorporates a better description of the computational approaches employed in the analysis of the sequencing data, which facilitates reproducibility and replicability of the results. The revised work constitutes a valuable contribution to the field of plant microbiota research.

However, there are few remaining suggestions for the authors to further improve the current version of the manuscript that could possibly be handled directly with the editorial office without the need for further revision.

Point 2. of the rebuttal letter:

The methods description has been substantially improved and including the scripts used for the analyses is not only important for clarity and reproducibility issues but also useful for other scientists. More extensive comments and documentation of the code and perhaps using a public repository instead of adding the commands to the supplement would be a further improvement.

It is encouraging that the main conclusions are consistent independently of the normalization scheme / rarefaction. However, I am surprised to read that the number of predicted core OTUs is larger when subsampling than normalization. I think this might be a typo in the rebuttal letter. In any case a smaller number of core OTUs seems more in line to previous studies.

I have two remaining comments concerning the analyses: first, why use PCA on transformed data instead of PCoA on pairwise distances (e.g. B-C or UniFrac)? The latter seems to be a more appropriate and far more common approach to use on microbial composition data. Also, Supplementary Fig. 5 and 6 would be appropriate as main figures (again, perhaps PCoA instead) as they give an intuitive view of the entire dataset, using normalized or subsampled data. It might also be interesting to have a parallel RDA plot using root instead of soil samples (and to use the same shapes consistently for compartments across figures).

Minor point 4.:

Indeed, what I meant was that there was a misspelling in the text, which is still there (now in line 132) "Firstly, we used principle component analysis (PCA) to summarize...".

Response to reviewers' comments

Reviewer #2 (Remarks to the Author):

[No further comments for author.]

Reviewer #3 (Remarks to the Author):

Revised manuscript by Yeoh, et al., "Evolutionary conservation of a core root microbiome across plant phyla along a tropical soil chronosequence".

The authors have adequately addressed my major concerns, including the question of how generalizable these findings are in light of the acidic soils of the chronosequence. The revised version of the manuscript now incorporates a better description of the computational approaches employed in the analysis of the sequencing data, which facilitates reproducibility and replicability of the results. The revised work constitutes a valuable contribution to the field of plant microbiota research.

However, there are few remaining suggestions for the authors to further improve the current version of the manuscript that could possibly be handled directly with the editorial office without the need for further revision.

Point 2. of the rebuttal letter:

The methods description has been substantially improved and including the scripts used for the analyses is not only important for clarity and reproducibility issues but also useful for other scientists. More extensive comments and documentation of the code and perhaps using a public repository instead of adding the commands to the supplement would be a further improvement.

We are willing to provide this documentation in a public repository if the editor wants us to do so, however, this information is now added in supplementary material which should be adequate for readers to reproduce our results.

It is encouraging that the main conclusions are consistent independently of the normalization scheme / rarefaction. However, I am surprised to read that the number of predicted core OTUs is larger when subsampling than normalization. I think this might be a typo in the rebuttal letter. In any case a smaller number of core OTUs seems more in line to previous studies.

The number of core OTUs derived from total sum-scaled counts indeed is lower than from subsampled data. We think that since soils and plant rhizosphere communities are complex, subsampling may have missed rarer OTUs in soils, present at higher relative abundances in roots, creating the impression that they are root-specific. Since the inclusion of an OTU as part of the

“core” is somewhat dependent on the analytical method used, we summarized our data at the genus level and found a high degree of consistency between the two methods at this taxonomic resolution (Suppl. Tables 3 and 7).

I have two remaining comments concerning the analyses: first, why use PCA on transformed data instead of PCoA on pairwise distances (e.g. B-C or UniFrac)? The latter seems to be a more appropriate and far more common approach to use on microbial composition data.

We chose to perform PCA because this method preserves original values/distances between the centered log ratio-transformed microbial abundance data. Bray-Curtis or UniFrac transformation on centered log ratio-transformed data is not a valid operation because the transformed data contains negative values.

Also, Supplementary Fig. 5 and 6 would be appropriate as main figures (again, perhaps PCoA instead) as they give an intuitive view of the entire dataset, using normalized or subsampled data. It might also be interesting to have a parallel RDA plot using root instead of soil samples (and to use the same shapes consistently for compartments across figures).

A principal component ordination of root samples colour-coded by plant community and host lineage is now included as Supplementary figure 8.

Minor point 4.:

Indeed, what I meant was that there was a misspelling in the text, which is still there (now in line 132) “Firstly, we used principle component analysis (PCA) to summarize...”.

We thank the reviewer for pointing this out- the spelling mistake has been corrected.